# A location-based service scheme with attribute information privacy

**Zhiguo Dai** [ID]*, **Jichao Li**

College of Information Science and Electronic Technology, Jiamusi University, Jiamusi, China

* dzg0454123@163.com

**Data Availability Statement:** The availability data is GeoLife GPS Trajectories which gets from the following URL. https://www.microsoft.com/en-us/research/publication/geolife-gps-trajectory-dataset-user-guide/.

**Funding:** This study was supported by the Basic Scientifics Research Operating Expenses of

## Abstract

In location-based service (LBS), private information retrieval (PIR) is an efficient strategy used for preserving personal privacy. However, schemes with traditional strategy that constructed by information indexing are usually denounced by its processing time and ineffective in preserving the attribute privacy of the user. Thus, in order to cope with above two weaknesses, in this paper, based on the conception of ciphertext policy attribute-based encryption (CP-ABE), a PIR scheme based on CP-ABE is proposed for preserving the personal privacy in LBS (location privacy preservation scheme with CP-ABE based PIR, short for LPPCAP). In this scheme, query and feedback are encrypted with security two-parties calculation by the user and the LBS server, so as not to violate any personal privacy and decrease the processing time in encrypting the retrieved information. In addition, this scheme can also preserve the attribute privacy of users such as the query frequency as well as the moving manner. At last, we analyzed the availability and the privacy of the proposed scheme, and then several groups of comparison experiment are given, so that the effectiveness and the usability of proposed scheme can be verified theoretically, practically, and the quality of service is also preserved.

## 1 Introduction

In current, along with the development of wireless communication and position technology, location-based service (LBS) becomes more and more popular in people's daily life. However, as this type of service must have the location of the user in advance to prepare the feedback, more and more people begin to pay close attention to the problem of violating personal privacy [1,2]. In general, schemes of privacy preservation can be briefly classified into two categories: the strategy of generalization [3] as well as the strategy of obfuscation [4]. However, these schemes had to confront the same problem, as during the process of preparing feedback, it is unavoidable that the LBS server must learn the purpose of the user, so as to find the result and feedback it to the user. During this procedure, the personal privacy (such as the type of point of interests (PoI), the query types and so on) will be gained by an un-trusted LBS server by inferring the purpose, so these schemes cannot preserve the personal privacy effectively. Private information retrieval (PIR) can effectively cope with this problem [5]. In PIR, the LBS server encrypted the PoIs that stored in its database, and then with the comparison of binary

Heilongjiang Provincial University and Colleges
(Grant No. 2021-KYYWF-0581). The funder took
role in study design and preparation of the
manuscript, but had no role in data collection and
analysis, or decision to publish.

**Competing interests:** The authors have declared
that no competing interests exist.

index or hardware index to get feedback result without violating any privacy to any entity. Then based on the differential of longitude and latitude in geography, Wightman et al. [6] proposed a mapping based PIR. Yang et al. [7] optimized the calculable PIR with a trusted central server, so the speed of blind query disposing can be accelerated.

In spite of this, as a feature of LBS is the PoIs must be feeding back in real time, but the whole process of encryption and comparing in PIR needs a large amount of time. In addition, in order to preserve the privacy, the LBS server also needs to index more results to keep the privacy, so most times PIR will affect the quality of service in LBS [8]. Based on the conception of increasing the index efficiency, Hu et al. [9] proposed a hierarchical index structure and Yi et al. [10] proposed a fuzzy index structure for PIR. However, these structures did not consider the attributes of a user can also be used in identifying the privacy. Furthermore, in LBS, the location privacy can also be inferred by some attributes, such as the velocity of moving, the query interval and so on [11]. For preserving the attribute privacy, and at the same time reduce the cost of calculation and increase the quality of service, we consider utilizing the generalized attributes as an index and compared the encryption set to retrieve the feedback. Then based on above conception, we proposed a LPPCAP scheme that used in LBS to preserve the personal privacy. In this scheme, the query set is constructed by the set of attributes and the process of comparison also based on this set, so the process of retrieving will be simplified. In addition, the attributes are also encrypted to preserve the attribute privacy. Though above two aspects and compared with traditional PIR, the superiority of this scheme can be reflected in both privacy preservation and quality of service. At last, performance analysis as well as simulation experiment is given, so that the result will further demonstrate the superiority of the proposed scheme. The contribution of this paper can be summarized as the following three points.

1. We proposed a LPPCAP scheme, which can preserve the personal privacy for using location based service, and without any information be leaked to other entities.

2. We utilized the set of attributes and the process of comparison to retrieve the query result, so as to simplify the process of retrieving back the requiring result with privacy preservation, and further strengthen the resist ability of LPPCAP which makes the maximum uncertainty for the adversary to identify the user.

3. We conduct comprehensive experiments on efficiency and utility and with the results the superiority of the proposed scheme is demonstrated, then the results show the level of privacy preservation and the quality of service is better than other schemes.

The organizational structure of this paper can be briefly summarized as follows. In Section 1, we analyze related works such as anonymity and encryption in location privacy preservation. Section 2 shows the system environment and the requirement of privacy preservation. In Section 3, the specific strategy and workflow of LPPCAP is shown. Section 4 shows the experimental settings, comparison results as well as the reason for these results. At last, we conclude this work and analyze future works in Section 5.

## 2 Related works

In location-based privacy protection processing, existing privacy protection methods can be simply classified into two main strategies: anonymous and encryption.

Anonymous privacy protection is mainly divided into two categories: $k$-anonymity [12] and $\varepsilon$-indistinguishability [13]. The former finds at least $k$ similar users through a central server [14] or user cooperation [15], and submits the information of all k users to the location service provider to disrupt the attacker's accurate identification of the user, thus protecting the

user's personal privacy through the attacker's misidentification. In the central server method, privacy protection is mainly achieved by $k$-anonymity generalization of user movement positions [16], online collaborative $k$-anonymity privacy protection for cloud services [17], and anonymous allocation of multiple tasks for differential privacy protection [18]. In the user cooperation method, current research focuses more on building collaborative anonymous groups through blockchain [19], providing effective feedback under anonymous collaboration [20], and privacy protection for multi-cooperative user perception under anonymity [21].

In privacy protection methods mainly based on differential privacy, various noises satisfying differential privacy are mainly added to achieve the indistinguishability between user sensitive information and other information [22]. Current research results mainly include road network indistinguishable algorithms for privacy protection in road environments [23], 3D geographic indistinguishable algorithms for indoor environments [24], spatial crowdsourcing indistinguishable algorithms for vehicle network crowd sensing [25], and personalized local location indistinguishable algorithms for user differences [26] etc.

Undoubtedly, in addition to the above two main applications, there are other similar strategies that adopt different strategies depending on the focus of privacy protection, such as multi-anonymous privacy protection methods for online ride-hailing privacy protection [27], semantic privacy protection anonymity for data sharing [28], and privacy protection methods for inadvertent sharing of indoor privacy navigation [29]. Liu et al. [30] also proposed a fully-distributed context-aware trust model for location based service. These methods further enrich the application environment and scope of anonymous privacy protection strategies.

Based on encryption techniques, information can be hidden to make it more difficult for attackers to obtain user personal information, thereby providing effective privacy protection [1]. According to the differences in the use of encryption methods, they can be divided into encryption comparison and privacy information retrieval [31]. In the encryption comparison, the main strategy is to compare two encrypted quantities with each other, so as to complete the privacy protection task or location matching without obtaining any additional information between each other, such as the parking lot allocation algorithm for privacy-protected path matching [32], geographic range query matching algorithm for mobile crowdsensing, and access authorization for edge computing [33]. In privacy information retrieval, decentralized asynchronous retrieval feedback [34], privacy information retrieval for semantic information [35], and intermittent privacy information retrieval in location privacy processing [36] are mainly used. For schemes utilize the conception of PIR, Vithana et al. [35] utilized semantic as the index to generalize the user with similar semantic to preserve the privacy of the user. Then based on the differential of longitude and latitude in geography, Wightman et al. [6] proposed a mapping based PIR. Yang et al. [7] optimized the calculable PIR with a trusted central server, so the speed of blind query disposing can be accelerated. For schemes utilize the conception of CP-ABE, Li et al. [37] utilized adversarial attacks to protect personal attribute privacy. Huang et al. [38] utilized ABE to hidden policy in cloud services. Lai et al. [8] proposed a scheme with CP-ABE to achieve PIR.

Since classify of the privacy protection scheme of generalization is always a traceable way, there will always be a risk probability of producing location or location sets, so there is still a high privacy risk when the attacker has sufficient background knowledge. In the privacy protection strategy using encryption techniques, the comparison method in location privacy is difficult to implement throughout the entire service stage, so the PIR strategy that can complete information retrieval in a dense environment without displaying any plaintext user information is more practical and has better privacy protection effectiveness in location privacy protection.

## 3 Preliminaries

### 3.1 System architecture and privacy threat

In general, there are two types of system architectures used for preserving privacy in LBS: they are centralized architecture and distributed architecture. The centralized architecture usually employs a trusted central server that disposes the generalization or obfuscation, so the central server may become the attack focus or service bottleneck. The distributed architecture usually utilizes the mobile device and collaborates with other users or generates a generalized query set to conceal the real intention. As PIR is a scheme that whole process of query and feedback is encrypted, it does not need any central server or collaborative users, so the system architecture is a distributed architecture and the detail structure can be seen in Fig 1.

From Fig 1, two entities can be seen and they are called the user as well as the LBS server. The user is the moving user equipped with location and communication device, so that he/she can send the service request to a LBS provider and gets the result. The LBS server is the service provider. This entity usually gets the request from a user and finds the result from historical data then feedback it to the user with service module. In most times, the LBS server is usually seen as a trusted entity, as it is usually established by the government or large enterprises. However, as the LBS server stores mass of personal data, it may become the attack focus and once breached by an adversary, and then the personal privacy will be violated. In addition, in case of attract by some huge commercial interests, the enterprise may also violate the privacy. Thus, in this paper, we assume the LBS server is a semi-trusted entity, as it may be curious about the privacy of the user, but can abide the agreement and find the result from historical data and feeds back the query result to the user.

Based on architecture mentioned above, the process of LBS can be depicted as the following. First, the user sends <id, location, query> to the LBS server and requires for the result (such as where is the nearest restaurant or the oil station along our journey). Secondly, the LBS server finds the result with <id, location, query>, and feeds back the result to the user. During this process the LBS server will get the privacy (such as location, query) of the user. In a classic model of $k$-anonymity, the location, the query is generalized with other locations, queries, so as the LBS server cannot distinguish which is the precise location or query of the user, and then the probability of guessing the precise user is $p(id{\rightarrow}l,q) = 1/k$. However, as LBS server can get some background knowledge to make the probability of guessing the precise user become $p(id{\rightarrow}l,q|b){\ll}1/k$. In addition, as the probability of guessing satisfy $p(id_1{\rightarrow}l,q|b){\neq}p(id_2{\rightarrow}l,q|b)$ in these $k$ users, the LBS server will be easier to get the privacy.

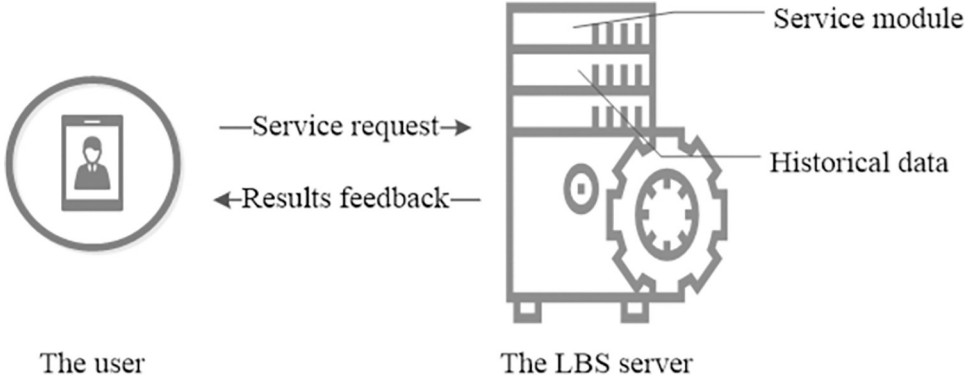

**Fig 1. System architecture of LPPCAP.**

## 3.2 Requirements for privacy preservation

According to the threat introduced in section 3.1, we consider the LBS server as the adversary, so in order to preserve the personal privacy an efficient scheme must satisfy the following conditions.

1. During the process of getting LBS result, there will be less or nearly no personal information published to the LBS server.

2. Without any precise information about the query, the LBS server has the ability to find out the result and feedback it to the user.

3. The LBS server cannot identify any special attributes from the query request and the id has the same probability to correlate to others, so as
$p(id_1 \rightarrow l, q|b) = p(id_2 \rightarrow l, q|b) = \ldots = p(id_k \rightarrow l, q|b).$

4. The feedback result must be sent to the user in an endurable time.

Thus, based on above requirements and with the help of CP-ABE and PIR conception, in this paper, we propose a LPPCAP scheme to preserve the personal privacy of the user.

## 3.3 The conception of LPPCAP

In general, if a user wants to use the LBS, he/she must send a query to the LBS server, and then the LBS server finds result with the query and sends back the result to the user, the query may be "where is the nearest restaurant", "find the shortest path to gas station", "show me the service point every 5 minutes" and so on. So the query can be formalized as $Q = \{(x,y),t,c\}$, where $(x,y)$ denotes the current location of the user, $t$ is the query time and $c$ is the content of query. If we see these elements as attributes, then the query can be seen as $A = \{A_1,A_2,\ldots,A_n\}$, where $A_i, 0 \leq i \leq n$ denotes an element used in query. As a result, the set of attributes can be used to retrieve the feedback result. In addition, if this set is obfuscate with other similar sets and encrypted by an encryption scheme, it will be more secure than other schemes. Meanwhile, the consistency of encrypted attributes will facilitate the process of finding the result in the LBS server. Thus, based on these ideas, the brief process of LPPCAP can be summarized as shown in Fig 2.

# 4 LPPCAP

## 4.1 The parameters used in LPPCAP

According to the conception of LPPCAP, the whole process can be divided into two parts: the part of disposing attribute by the user and the part of disposing result retrieving by the LBS server. In order to facilitate the comprehension of disposing PIR, the parameters used in LPPCAP are shown in Table 1. Then based on the execution sequence of PIR proposed in literature [8], a detailed introduction on how the requirement changed into the feedback result is shown in follows.

## 4.2 The process of LPPCAP

Before sending the requirement, the user as well as the LBS server must predispose the query information and the PoIs stored in the server. For the LBS server, the whole data of PoIs stored in this server must be encrypted with $F(1^\lambda) \rightarrow (\mathbb{G}, p, g), \alpha \xleftarrow{\$} \mathbb{Z}_p, g_1 \leftarrow g^\alpha$. Then the LBS server publishes the public parameter $params = (g, g_1, \mathbb{G})$. Where $g$ is the generator of $\mathbb{G}$, $\mathbb{G}$ is a $p$ order large prime cyclic group, $F(\cdot)$ is a polynomial time probability algorithm. The key generation algorithm is given in algorithm 1.

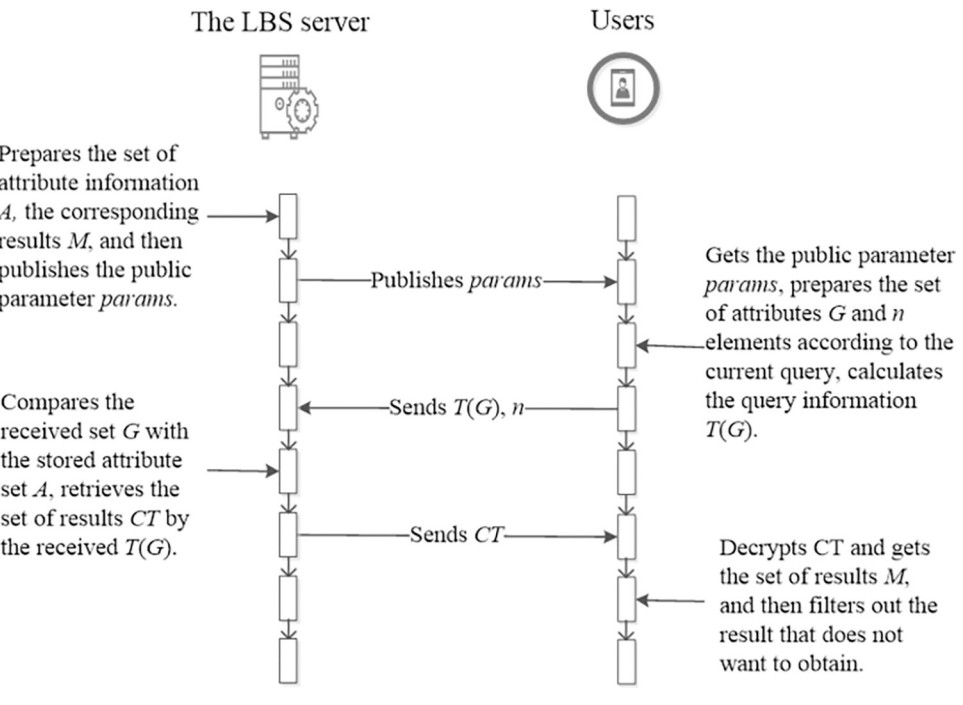

**Fig 2. The protocol of LPPCAP.**

**Table 1. Parameters used in LPPCAP.**

| Notation | Description |
|---|---|
| $A$ | The set of attributes |
| $\lambda$ | The system secure parameter selected by the LBS server |
| $params$ | The published parameter by the LBS server |
| $G$ | The set of attributes that selected by the user |
| $\mathbb{Z}_p^n$ | The set that contains at least $n$ elements selected from $\mathbb{Z}_p$ |
| $\mathbb{Z}_p$ | The set that constructed by number of $0 \sim p-1$ |
| $sk$ | The private key of the user |
| $\beta$ | The private key stored by the user |
| $\leftarrow^\$$ | Consistently select an element from the right set and assign its value to the left set |
| $T(G)$ | The encrypted query that sent to the LBS server |
| $M$ | The result that the user needs |
| $\mathbb{A}$ | The set of attributes that the LBS server stored |
| $k$ | The number of attributes that the LBS server stored |
| $M'$ | The encrypted information sent to the user |
| $n$ | The number of attributes that the user selected |
| $CT$ | The set of feedback results sent by the LBS server |

```
Algorithm 1. The key generation algorithm.
Input: $\mathbb{G}, p, g, \lambda$
Output: the public parameter $params = (g, g_1, \mathbb{G})$
1 The LBS server chooses
2 $\alpha \leftarrow^{\$} \mathbb{Z}_p$
3 $g_1 \leftarrow g^{\alpha}$
4 $F(1^{\lambda}) \rightarrow (\mathbb{G}, p, g)$
5 Return $(g, g_1, \mathbb{G})$
```

In the process of disposing PoIs, suppose that all attributes of PoIs can be denoted as $A$. The LBS server has to choose the secure parameter $\lambda$, the order of prime number $p$ as well as the public parameter $params$ in advance. At the same time, for the user, he/she has to generate the set of selected attributes with the private key, and sends the encrypted set to the LBS server. In order to get the encrypted set of selected attributes, the user has to complete the following operations. Suppose that the selected attributes of the user can be denoted as $G = \{A_1, A_2, \ldots, A_n\} \in \mathbb{Z}_p^n$, where $n$ is the number of attributes that the user selected. With the set of attributes $G = \{A_1, A_2, \ldots, A_n\} \subset A$ as well as a random number $\beta \leftarrow \mathbb{Z}_p$, the user calculates the following parameters.

$$h \leftarrow g^{\beta}$$

$$\overrightarrow{t} = (t_1, t_2, \ldots, t_n) \leftarrow^{\$} \mathbb{Z}_p^n$$

$$\overrightarrow{s} = (s_1, s_2, \ldots, s_n) \leftarrow^{\$} \mathbb{Z}_p^n$$

$$\overrightarrow{r} = (r_1, r_2, \ldots, r_n) \leftarrow^{\$} \mathbb{Z}_p^n, g_1 \neq g^{-\beta r_i}, i = 1, 2, \ldots, n$$

Then for each $i$ the calculates:

$$u_i \leftarrow g_1 h^{r_i}, v_i \leftarrow g^{r_i}, X_i \leftarrow g^{s_i}, Y_i \leftarrow g^{t_i}$$

$$U_i \leftarrow h^{s_i} \prod_{j=1}^{n} u_j^{a_{j,i-1}}$$

$$V_i \leftarrow h^{t_i} \prod_{j=1}^{n} v_j^{a_{j,i-1}}$$

With parameters calculated by above operations, the user sends the encrypted attribute information $T(G) = \{U_i, V_i, X_i, Y_i\}$, $i = 1, 2, \ldots, n$ to the LBS server, and at the same time preserve the private key $\beta$. Where $a \leftarrow^{\$} b$ denotes the process of selecting an element from the set of $a$ and assigns it to the set $b$. We elaborate above process in algorithm 2.

```
Algorithm 2. The query encryption.
Input: The set of attributes converted from user's query G and the
private key $\beta \leftarrow \mathbb{Z}_p$
Output: The encrypted query information T(G)
1 Randomly choose the initial parameters $\overrightarrow{t}$, $\overrightarrow{s}$, $\overrightarrow{r}$;
2 Calculates $h \leftarrow g^{\beta}$;
3 for(i = 1, i <= n, i++)
4   $u_i \leftarrow g_1 h^{r_i}$, $v_i \leftarrow g^{r_i}$;
5   $X_i \leftarrow g^{s_i}$, $Y_i \leftarrow g^{t_i}$;
6   $U_i \leftarrow h^{s_i} \prod_{j=1}^{n} u_j^{a_{j,i-1}}$;
7   $V_i \leftarrow h^{t_i} \prod_{j=1}^{n} v_j^{a_{j,i-1}}$;
8 end
9 return $T(G) = \{U_i, V_i, X_i, Y_i\}$;
```

In algorithm 1, line 3–9 shows the process of encrypting each attribute that corresponds to the query for LBS. During the process of getting the encrypted information $T(G)$, the time complexity seems equal to $O(n)$, but as the multiplicative calculation in lines 6–8, the real time complexity may be $O(n^2)$.

Once the LBS server receives the encrypted query $T(G)$ from the user, for the data set of PoIs $M$ as well as the attributes set $\mathbb{A}$, $|\mathbb{A}| = k$, the LBS server has to calculate the following parameters according to the attribute set $\mathbb{A}$.

$$P_i \leftarrow V_1 \cdot V_2^{A_i} \cdot V_3^{A_i^2} \ldots V_n^{A_i^{n-1}} = v_i \cdot h^{t_1 + t_2 A_i + \ldots + t_n A_i^{n-1}} = g^{r_i} \cdot h^{\eta_i}$$

$$Q_i \leftarrow U_1 \cdot U_2^{A_i} \cdot U_3^{A_i^2} \ldots U_n^{A_i^{n-1}} = u_i \cdot h^{s_1 + s_2 A_i + \ldots + s_n A_i^{n-1}} = g_1 g^{\beta r_i} \cdot h^{\theta_i} \; W_i \leftarrow Q_i \cdot g_1^{-1} = g^{\beta r_i} h^{\theta_i}$$

At the same time the LBS server calculates:

$$l_1, l_2, \ldots, l_k \leftarrow^{\$} \mathbb{Z}_p$$

$$P \leftarrow \prod_{i \in \mathbb{A}} P_i^{l_i} = g^{\sum_{i \in \mathbb{A}} r_i l_i} \cdot h^{\sum_{i \in \mathbb{A}} \eta_i l_i}$$

$$W \leftarrow \prod_{i \in \mathbb{A}} W_i^{l_i} = g^{\sum_{i \in \mathbb{A}} \beta r_i l_i} \cdot h^{\sum_{i \in \mathbb{A}} \theta_i l_i}$$

$$t \leftarrow^{\$} \mathbb{Z}_p$$

$$C_0 = \prod_{i \in \mathbb{A}} (\prod_{j=1}^{n} X_j^{A_i^{j-1}})^{t l_i} = g^{t \cdot \sum_{i \in \mathbb{A}} \theta_i l_i}, \theta_i = s_1 + s_2 A_i + \ldots + s_n A_i^{n-1}$$

$$C_1 = \prod_{i \in \mathbb{A}} (\prod_{j=1}^{n} Y_j^{A_i^{j-1}})^{t l_i} = g^{t \cdot \sum_{i \in \mathbb{A}} \eta_i l_i}, \eta_i = t_1 + t_2 A_i + \ldots + t_n A_i^{n-1}$$

$$C_2 = P^t$$

$$C_3 = W^t \cdot M$$

With parameters calculated by above operations, the LBS server gets the encrypted result set $CT = (C_0, C_1, C_2, C_3)$ and sends this set to the user, the whole process of preparing the encrypted result is shown in algorithm 3.

```
Algorithm 3. The process of information retrieval in the LBS server.
Input: The encrypted information from the user T(G)
Output: The set of encrypted results CT established by the LBS server
1 for(i=1,i<=k,i++)
2     Calculates P_i,Q_i,W_i;
3 end
4 Randomly chooses l_1~l_k,t;
5 Repeat calculates the result of P×W;
6 Calculates C_0,C_1,C_2,C_3;
7 return CT = (C_0,C_1,C_2,C_3)
```

In algorithm 3, all probable PoIs is selected with the encrypt attributes sent by the user. In addition, as the scale of encrypted attributes is much higher than the user $k \gg n$, the time complexity of algorithm 3 is $O(k) + O(n) = O(k)$.

The user has to decrypt the plaintext from the encrypted set of results $CT$ sent from the LBS server with the following calculation.

$$\begin{aligned}
M' &= (C_1^{-sk} \cdot C_2)^{-sk} \cdot C_0^{-sk} \cdot C_3 \\
&= ((g^{t \cdot \sum_{i \in \mathbb{A}} \eta_i l_i})^{-\beta} \cdot (g^{t \cdot \sum_{i \in \mathbb{A}} r_i l_i} \cdot h^{t \cdot \sum_{i \in \mathbb{A}} \eta_i l_i}))^{-\beta} \cdot \\
&\quad (g^{t \cdot \sum_{i \in \mathbb{A}} \theta_i l_i})^{-\beta} \cdot g^{t \cdot \sum_{i \in \mathbb{A}} \beta r_i l_i} \cdot h^{t \cdot \sum_{i \in \mathbb{A}} \theta_i l_i} \cdot M \\
&= M
\end{aligned}$$

At last, the user refines the real result that he/she needs. The process of the user decrypts the plaintext is shown in Algorithm 4.

```
Algorithm 4. Decrypt the set of feedback results.
Input: The encrypted results set CT that sends from the LBS server,
g,g_1,G,sk
Output: The plaintext M'
```

1  $M' = (C_1^{-sk} \cdot C_2)^{-sk} \cdot C_0^{-sk} \cdot C_3;$

   $M' = (C_1^{-sk} \cdot C_2)^{-sk} \cdot C_0^{-sk} \cdot C_3$

   $= ((g^{t \cdot \sum_{i \in \mathbb{A}} \eta_i l_i})^{-\beta} \cdot (g^{t \cdot \sum_{i \in \mathbb{A}} r_i l_i} \cdot h^{t \cdot \sum_{i \in \mathbb{A}} \eta_i l_i}))^{-\beta} \cdot$

   $(g^{t \cdot \sum_{i \in \mathbb{A}} \theta_i l_i})^{-\beta} \cdot g^{t \cdot \sum_{i \in \mathbb{A}} \beta r_i l_i} \cdot h^{t \cdot \sum_{i \in \mathbb{A}} \theta_i l_i} \cdot M$

   $= M$

3  Return $M'$

In algorithm 4, each attribute is calculated by iterate addition, so as to get the plaintext of query result, as a result the time complexity of algorithm 4 is $O(n)$.

With three algorithms mentioned in above, the process of LPPCAP is shown, and the user can utilize this scheme to get the service without leaking any private information.

# 5 Performance evaluation

## 5.1 Security and availability analysis

The security of LPPCAP depends on the security of attribute encryption scheme that is mentioned in literature [8], so we do not focus the features of encryption. In addition, LPPCAP can be seen as a practical of the theory produced in literature [8]. In security analysis, we mainly focus on the difficulty of adversary identifies the real attributes. In availability analysis, we focus on the accuracy of feedback as well as the running time in security retrieval.

In security, as the user's attributes are encrypted by $A \bmod p$, and $p$ is a prime that large enough. So according to the character of modular arithmetic, although the adversary gets the number of $p$, with the parameter $\beta$ he/she still difficult to infer the real attributes by inverse calculation. Then, suppose that, if the adversary has obtained the encryption information $T(G)$ and the number of attributes $n$, without $\overrightarrow{t}$, $\overrightarrow{s}$ and $\overrightarrow{r}$, the adversary still difficult to infer the real attributes, in spite of this information does not disposed by modular arithmetic, because $\overrightarrow{t}$, $\overrightarrow{s}$ and $\overrightarrow{r}$ are selected consistent with each other. At last, as the number of attributes that the LBS server stored is much higher than the user $k \gg n$, it will be difficult for the LBS server to identify the real set from the user and the success ratio of identifying the real set will be less than $1/k^n$, because the process of finding at least $n$ attributes sub-set from the set of at least $k$ attributes is much more difficult. In addition, as the attributes sent by the user may contain some attributes that do not belong to the user, so it will be even more difficult for the adversary identify the real attributes. In addition, as the adversary cannot decrypt the query of the user, the probability of guessing the real user for each location and query will be $p(id \rightarrow l, q|b) = 1/k$ and the adversary also difficult to distinguish the probability of $p(id_i \rightarrow l, q|b)$ and $p(id_j \rightarrow l, q|b)$ $i \neq j$, as $p(id_1 \rightarrow l, q|b) = p(id_2 \rightarrow l, q|b) = \ldots = p(id_k \rightarrow l, q|b)$. So if we utilize entropy to measure the privacy, we have $p(i) = p(id_i \rightarrow l, q|b)$, $H(i) = -\sum_{i=1}^{k} p(i) \log_2 p(i)$, the entropy $H(i)$ will get the maximum value, which means the adversary will have the maximum uncertainty of the user.

For the availability of proposed scheme, the accuracy of LPPCAP can be denoted as $params \leftarrow^{\$} F(1^\lambda)$, $(T(G), n, sk) \leftarrow^{\$} (params, G)$, $1 \leftarrow^{\$} (params, T(G), n)$ and $CT \leftarrow^{\$} (params, \mathbb{A}, T(G), M)$, for parameters $(params, CT, sk)$ the value of retrieving will be $\begin{cases} M & if\ \mathbb{A} \subseteq G \\ \bot & if\ \mathbb{A} \subseteq G \end{cases}$, so the result set retrieved by attributes must be included in the storage of the LBS server and the user can obtain the real result in this set. For the running time, according to the time complexity mentioned in section 2, with the result described under three

algorithms we have the running time is less than $O(n^2)+O(k)+O(n) = O(m^2)$, so it can be completed in binomial time, the detailed running time will be shown in the result of experiments.

## 5.2 Experiment preparation

Based on the analysis of security and availability in the above sub-section, we can conclude that the proposed LPPCAP has a better performance in both privacy preservation and the availability in theoretically. In this section, we will further verify the performance of our proposed scheme with several groups of simulation experiments, and the schemes used for comparison include that the intermittent PIR [36], the semantic PIR [35] the approximate PIR [10], the attribute encryption scheme [39] as well as the correlation indistinguishable scheme [14]. We utilize the central part of the Geilife data to simulate the user in LBS. Then the simulation experiments are deployed in a laptop with Intel core I7, 8GB memories and windows10 operation system, and we utilize Matlab R2017a as the instrument to verify the performance. In addition, the results are calculated at least 500 times and utilize the average result to construct the line charts.

## 5.3 Results with brief explanations

Table 2 shows the performance of various schemes in both privacy preservation and execution efficiency. From this table we can see the LPPCAP has a middle accuracy in feeding back result, and has a lower running time compared with other schemes. In addition, as all information used in this scheme is encrypted during the process of querying, LPPCAP also leads zero knowledge leakage as other PIR scheme. Then this scheme also has a better performance in concealing the attribute as well as the attributes un-correlation than other schemes.

Fig 3 shows the success ratio of an adversary identifies the real attribute, which increases along with the increasing number of attributes. In this figure, we can see that the success ratio of LPPCAP does not changed dramatically, as this scheme utilize the encrypted attributes as index to find the result. Furthermore, this scheme can also add some dummy attributes in the query set to obfuscate the real attributes. For other scheme with PIR, performances of the intermittent PIR, the semantic PIR as well as the approximate PIR are similar to LPPCAP but a bit higher, as these schemes also conceal the real attribute but fail to dispose the attributes correlation, so that the results are different with each other. For the scheme of attributes encryption, although this scheme encrypts the attributes of the user, it needs to send the real attribute to the LBS server, so the success ratio is higher than others. On the other hand, the scheme mainly defined with other collaborative users, the more attributes used the more difficult to find the collaborative user, and without these user this scheme will be failed to preserve the privacy. At last, the correlation indistinguishable is mainly designed to generalize the correlation of attributes but not conceal the attributes or conceal the attribute correlation, so the success ratio is the highest.

**Table 2. The comparison result with other similar schemes.**

| Scheme | Attributes conceal | Zero knowledge leakage | Attributes un-correlation | Accuracy | Running time |
|---|---|---|---|---|---|
| Intermittent PIR [36] | √ | √ | × | middle | high |
| Semantic PIR [35] | √ | √ | × | middle | middle |
| Approximate PIR [10] | √ | √ | × | middle | middle |
| Attributes encryption [39] | √ | × | √ | high | low |
| Correlation indistinguishable [14] | √ | × | √ | low | low |
| LPPCAP | √ | √ | √ | middle | low |

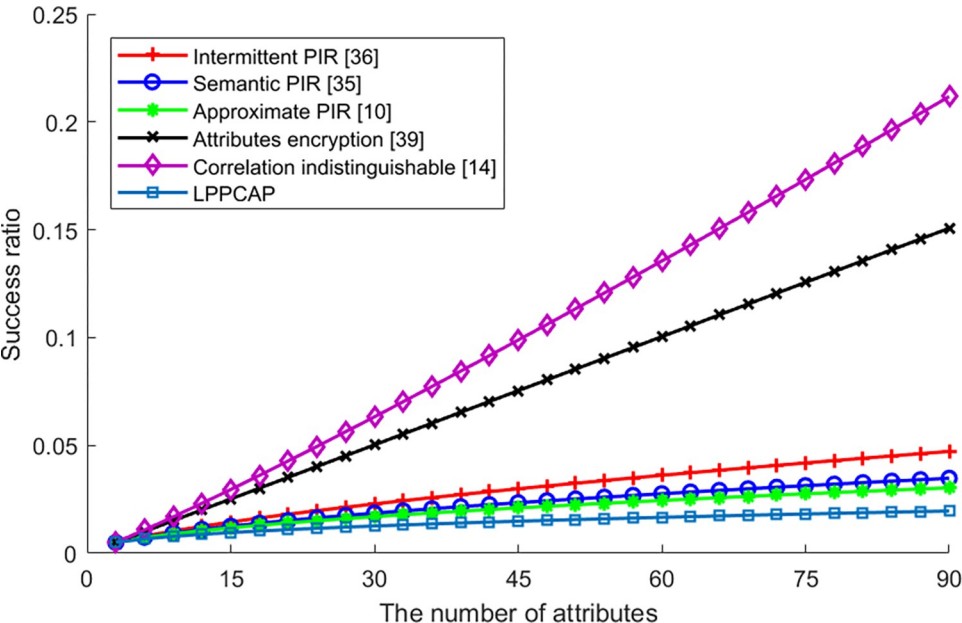

**Fig 3. The success ratio of identifying the privacy vs. the number of attributes.**

Fig 4 shows the success ratio of an adversary identifies the real attribute, which increases along with the increasing number of queries. From this figure, we can see that success ratios of all schemes are increasing with a higher number of queries, as more queries mean more information sent to the LBS server and more risk to be identified by the adversary. Among these schemes, LPPCAP performs best, as this scheme has the advantage of both attribute encryption as well as PIR, so the possibility of revealing the real attribute is the lowest. In addition, as the

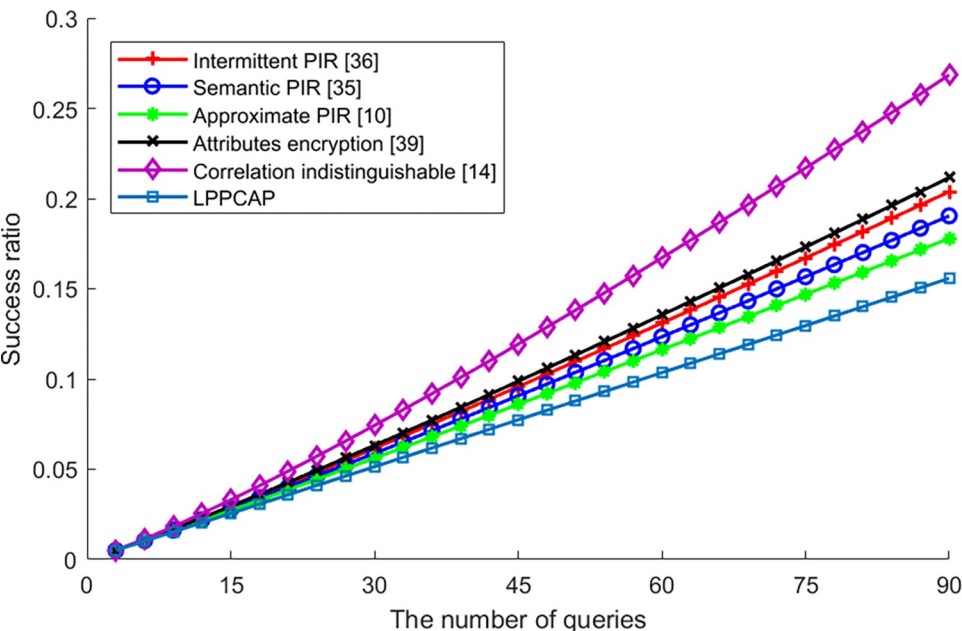

**Fig 4. The entropy of various algorithms vs. queries.**

correlation of attributes also obfuscated by dummies, the success ratio will be even lower. For other schemes with PIR strategy, such as the intermittent PIR, the semantic PIR as well as the approximate PIR, as they mainly utilize the encryption without disposing attribute to preserve the privacy, the success ratio is lower than the scheme without PIR, but higher than LPPCAP. Although the scheme of attributes encryption also encrypts the user's attributes, the collaborative users lack will affect the security and the success ratio of identifying will be higher. At last, the scheme of correlation indistinguishable that without encryption performs the worst.

Fig 5 shows the running time of various schemes changes along with attributes increasing. From this figure, we can see that the running time of all schemes is increasing with a higher number of attributes, as all schemes have to conceal or generalize each attribute, no matter of encryption or generalization, and the disposition for attribute occupied the running time. Among these schemes, the LPPCAP performs better than schemes with encryption, as this scheme utilizes the set of attributes as index, and can be used to compare with multidimensional data to find the results simultaneously, so the running time is shorter than others. But the running time of LPPCAP scheme is higher than the scheme of correlation indistinguishable, as the encryption costs much more time than selecting collaborative users. For schemes with encryption, as strategies of indexing are different from each other, the running time decreasing with the enhancement of index capability.

Fig 6 shows the running time of various schemes changes along with queries increasing. From this figure, we can see that the running time of all schemes is increasing with a higher number of queries, as more queries mean more sets of attributes have to be disposed, so the running time is higher. Among these schemes, the LPPCAP performs the best, as the running time in every query is shorter. However, the running time of scheme of correlation indistinguishable is higher than LPPCAP, as more queries mean more dummies added and longer distance shifted, so the running time is higher. For other schemes, such as the intermittent PIR, the semantic PIR as well as the approximate PIR, as strategies of indexing are different from each other, the running time also decreasing with the enhancement of index capability.

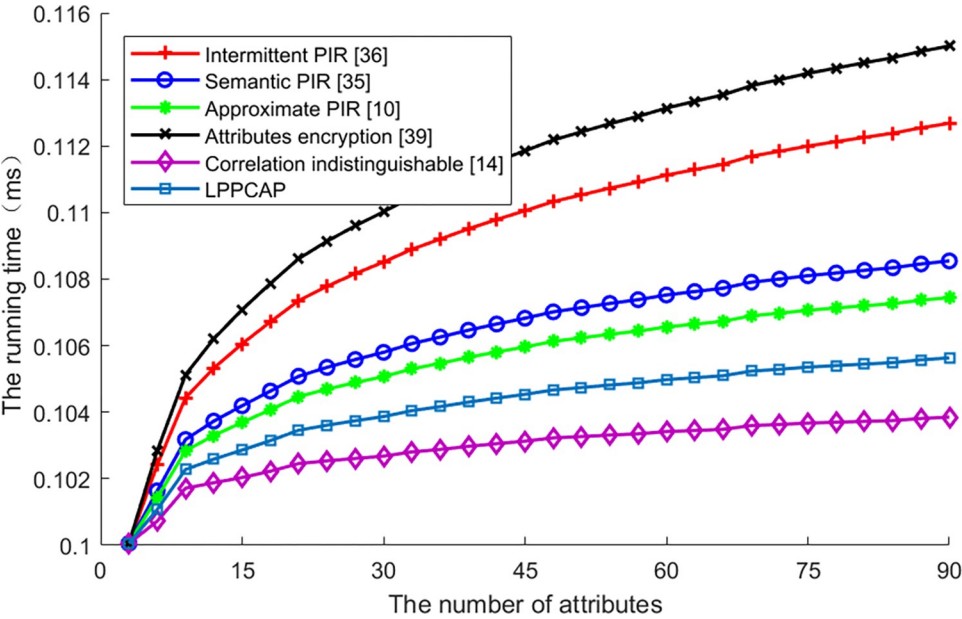

**Fig 5. The entropy of various algorithms vs. the number of attributes.**

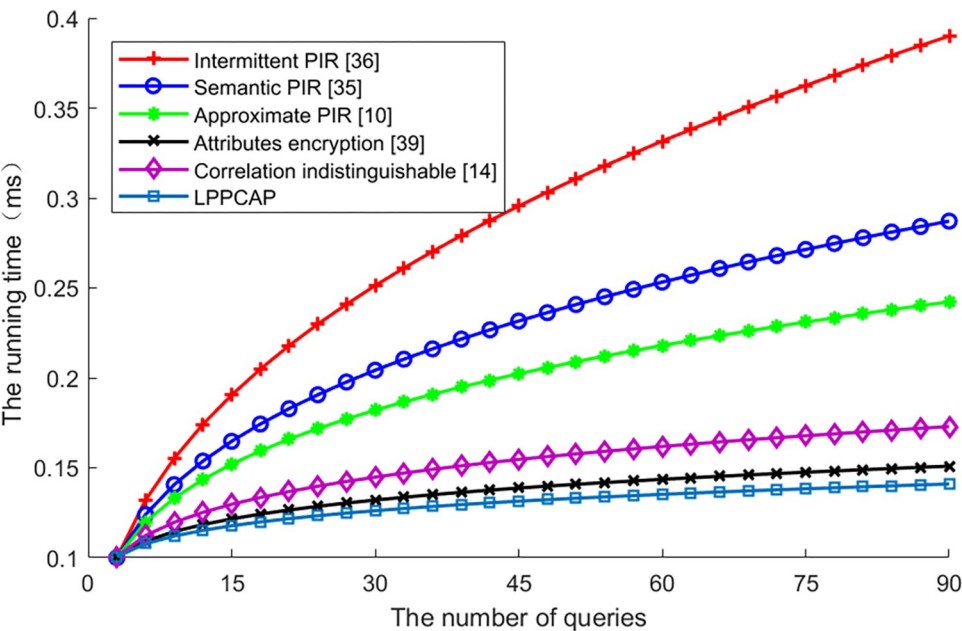

**Fig 6. The entropy of various algorithms vs. queries.**

Fig 7 shows the differences in algorithm success rates caused by changes in the number of user attributes during the execution process. It can be seen from the figure that privacy information retrieval algorithms (such as intermittent PIR, semantic PIR, approximate PIR, and the algorithm proposed in this paper) have relatively high success rates. This is mainly because these algorithms achieve privacy protection by encrypting their own information or attributes, without the need to find generalized users with similar attributes or features like generalization

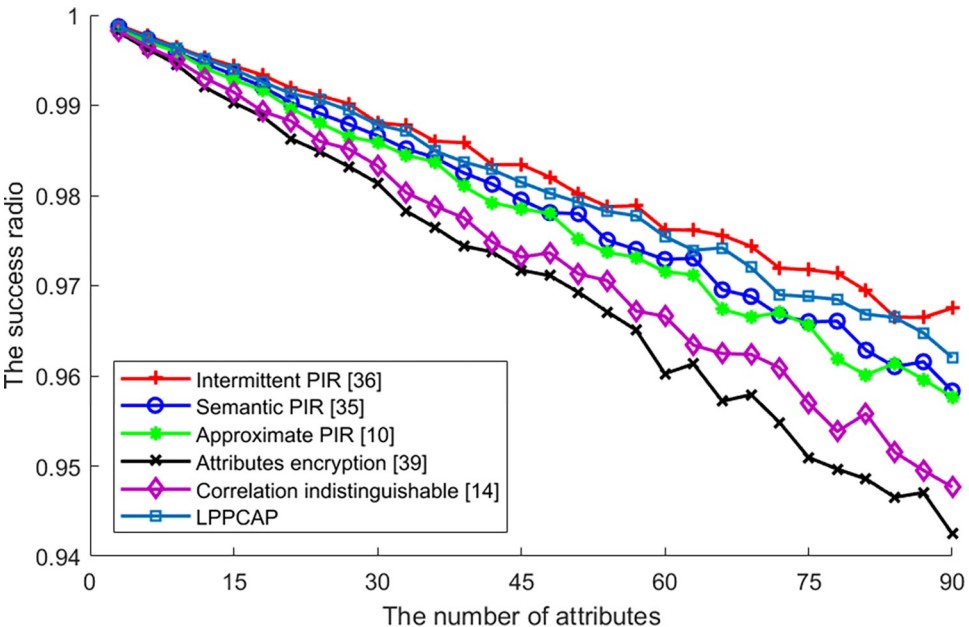

**Fig 7. The success ratio of various algorithms vs. the number of attributes.**

methods. Although the algorithm proposed in this chapter requires encryption of various attributes displayed by users, this processing does not cause significant fluctuations due to changes in the number of attributes. Therefore, like other PIR-based algorithms, the success rate of this algorithm is better than that of generalization algorithms. For the other two generalization algorithms participating in the comparison, both algorithms require finding generalized users with similar attributes to complete privacy protection processing for the applicant user. Therefore, as the number of attributes increases, the degree of reduction in the success rate of the algorithm is higher than that of PIR algorithms. In addition, the main reason for the reduction in the success rate of non-attribute encryption strategy algorithms in the above PIR algorithms with the increase in the number of attributes is similar to the success rate of attacks. The increase in the number of attributes increases the amount of background knowledge available to adversaries and more background knowledge directly allows attackers to guess, associate, and obtain user privacy information, leading to the failure of algorithm privacy protection. Therefore, these PIR algorithms will also experience a decrease in the success rate of algorithm execution due to an increase in the number of attributes.

Fig 8 shows the difference in algorithm success rates caused by changes in query frequency for different algorithms. It can be seen from the figure that algorithm query frequency is independent of algorithm success rate, meaning that the success rate of algorithm execution does not increase or decrease with changes in query frequency. This is mainly because each user query may or may not be successful. After averaging multiple queries, all unsuccessful executions are converted into an unsuccessful probability, which is not significant in overall comparison. Additionally, the success rate of generalization-based privacy protection algorithms (such as attribute-based encryption algorithms and probabilistic indistinguishability-based algorithms) is lower than that of PIR retrieval-based privacy protection algorithms. This is because generalization-based algorithms need to find generalized users who can generalize real users to complete privacy protection processing. If there are fewer generalized users in the current area or their willingness to participate in generalization is low, it is difficult for these

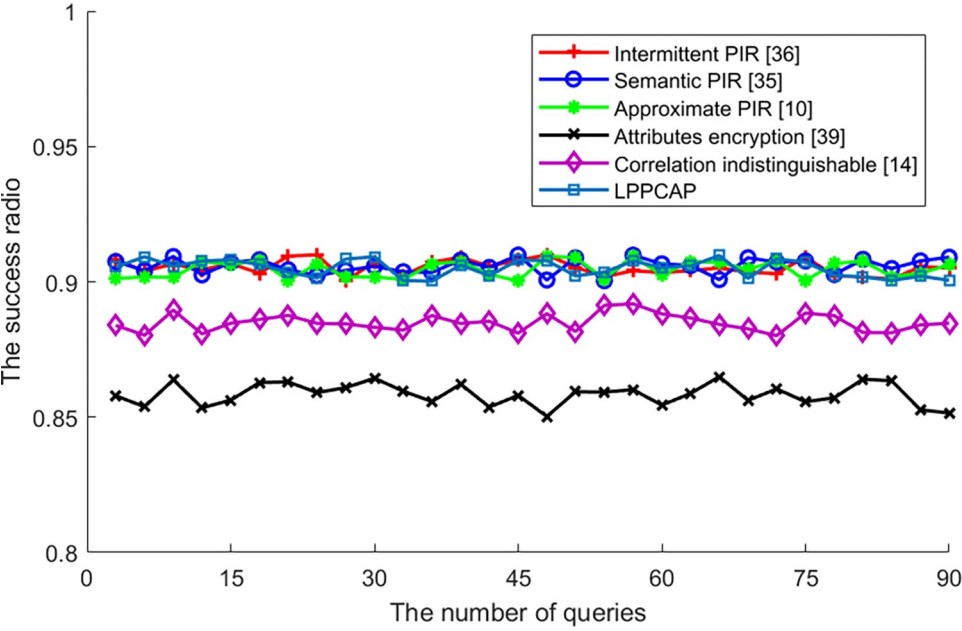

**Fig 8. The success ratio of various algorithms vs. Queries.**

algorithms to find enough generalized users, resulting in lower execution success rates for their privacy protection algorithms than for PIR retrieval-based privacy protection algorithms. Among many PIR retrieval-based privacy protection algorithms, the algorithm proposed in this article has little difference in algorithm execution success rate affected by query frequency compared with other algorithms. It can be regarded as having a high algorithm execution success rate along with changes in query frequency among similar algorithms.

In conclusion, we can consider that the LPPCAP has a better performance in both privacy preservation and the availability, no matter in theoretically or simulation experiment, and then the superiority of LPPCAP has been verified.

## 6 Conclusion

In this paper, we focus on the problem of reducing the running time as well as the problem of identifying the privacy with user's attributes in LBS, so we proposed a LPPCAP. This scheme utilizes the conception of CP-ABE and with the obfuscated attributes to provide privacy preservation for the user in LBS. In this scheme, three algorithms are proposed for the user, the LBS server as well as the decryption to complete the process of secure retrieval. Furthermore, as the index with attribute is much easier, the running time of the proposed scheme is much lower than others. At last, we analyze security as well as availability of the proposed scheme, and then the results of simulation experiment further demonstrate the superiority.

## Supporting information

**S1 Data set.**
(ZIP)

## Acknowledgments

We would like to present our thanks to anonymous reviewers for their helpful suggestions.

## Author Contributions

**Data curation:** Jichao Li.

**Formal analysis:** Zhiguo Dai.

**Funding acquisition:** Zhiguo Dai.

**Methodology:** Zhiguo Dai.

**Resources:** Jichao Li.

**Software:** Zhiguo Dai.

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
