## [Decision Letter · Decision Letter 0]

30 Jul 2024

PONE-D-24-26572A location-based service scheme with attribute information privacyPLOS ONE

Dear Dr. Dai,

Thank you for submitting your manuscript to PLOS ONE. After careful consideration, we feel that it has merit but does not fully meet PLOS ONE’s publication criteria as it currently stands. Therefore, we invite you to submit a revised version of the manuscript that addresses the points raised during the review process.

We look forward to receiving your revised manuscript.

Kind regards,

Prof. Zhiquan Liu 

Academic Editor

PLOS ONE

Journal Requirements:

We would like to present our thanks to anonymous reviewers for their helpful suggestions. This work was supported by the Basic Research Project of Higher Education Institutions in Heilongjiang Province, Project Number: 2021-KYYWF-0581, Project Name: Research on Student Class Concentration Based on Multi feature Cascading and Multi task Learning.

5. Thank you for uploading your study's underlying data set. Unfortunately, the repository you have noted in your Data Availability statement does not qualify as an acceptable data repository according to PLOS's standards.

Reviewers' comments:

Reviewer's Responses to Questions

**Comments to the Author**

1. Is the manuscript technically sound, and do the data support the conclusions?

Reviewer #1: No

Reviewer #2: Partly

Reviewer #3: Partly

2. Has the statistical analysis been performed appropriately and rigorously? 

Reviewer #1: No

Reviewer #2: I Don't Know

Reviewer #3: Yes

3. Have the authors made all data underlying the findings in their manuscript fully available?

Reviewer #1: No

Reviewer #2: Yes

Reviewer #3: Yes

4. Is the manuscript presented in an intelligible fashion and written in standard English?

Reviewer #1: Yes

Reviewer #2: Yes

Reviewer #3: Yes

5. Review Comments to the Author

**Reviewer #1: **In this paper, the authors try to propose a CP-ABE based PIR by utilizing CP-ABE technology.

However, the following questions should be considered.

(1) Inadequate motivation and problem formulation: The introduction fails to convincingly motivate the need for a CP-ABE based PIR. The specific research problems and challenges that the paper aims to address are not clearly articulated. The claimed contributions are not well-positioned against prior works, making it hard to gauge their significance. A stronger case needs to be made for why the proposed CP-ABE based PIR is an important advancement over existing solutions.

(2) Lack of technical depth and insights: The proposed construction does not offer significant technical innovations beyond combining existing techniques from CP-ABE based PIR. The paper lacks new concepts, insights, or design principles that push forward the state-of-the-art in this domain.

(3) In this paper, the authors proposed a CP-ABE based PIR by utilizing CP-ABE technology. However, the authors do not give the related works about CP-ABE.

(4) The authors proposed a CP-ABE based PIR. However, they do not give the concrete name of the related algorithms.

(5) The proposed CP-ABE based PIR is not complete. Especially, key generation algorithm is not given. How can you encrypt and decrypt?

**Reviewer #2:** 1. It is not clear, what is the role of f(x) in algorithm 1?2. No threat model is presented in the paper. It will be good to include the threat model and present the security analysis accordingly.

3. It is not clear whether the LBS will know who is sending the query or not? If the LBS knows the individual who sends the query then privacy is lost.4. Decryption proof of correctness may be with all steps involved. It is not clear how the second step is derived from the first step.

5. Zero Knowledge conceal is used in table 2 however no where it is described.6. Figure 8 should be rechecked. X-axis is not clear. Is it time or the number of queries? Similarly figure 6.

6. The novelty should be clearly presented.

**Reviewer #3:** In this manuscript, in this paper, based on the conception of ciphertext policy attribute-based encryption (CP-ABE), a PIR scheme based on CP-ABE is proposed for preserving the personal privacy in LBS. Personal privacy in LBS is a topic worth studying. However, there are some issues to consider:

1. In the “Introduction”, the authors described that “the superiority of this scheme can be reflected in both privacy preservation and quality of service”. However, the quality of service is not represented in the scheme, if the quality of service is the innovation point, it should be described in detail in the scheme, and it needs to be reflected in the abstract and contribution.

2. The “3.1 The conception of CP-ABE based PIR” should describ in the “Preliminaries”, and the system architecture and privacy threats are more appropriate in the “CP-ABE based PIR”, the front of the scheme description.

3. What is the relationship between this scheme and literature 8, and what are the advantages of this scheme compared with literature 8?

6. PLOS authors have the option to publish the peer review history of their article (what does this mean?). If published, this will include your full peer review and any attached files.

Reviewer #1: No

Reviewer #2: No

Reviewer #3: No

---

## [Author Response · Author response to Decision Letter 0]

6 Aug 2024

Response to reviewer's comments

Reviewer #1: In this paper, the authors try to propose a CP-ABE based PIR by utilizing CP-ABE technology.

However, the following questions should be considered.

(1) Inadequate motivation and problem formulation: The introduction fails to convincingly motivate the need for a CP-ABE based PIR. The specific research problems and challenges that the paper aims to address are not clearly articulated. The claimed contributions are not well-positioned against prior works, making it hard to gauge their significance. A stronger case needs to be made for why the proposed CP-ABE based PIR is an important advancement over existing solutions.

We add the formulation for the threat model to further promote the innovation of the motivation, and add the proof for why the LPPCAP utilizes CP-ABE based PIR to preserve the privacy of user in LBS. In addition, in the introduction, we also add the motivation of why we utilize CP-ABE based PIR and the claim the contribution of this work.

(2) Lack of technical depth and insights: The proposed construction does not offer significant technical innovations beyond combining existing techniques from CP-ABE based PIR. The paper lacks new concepts, insights, or design principles that push forward the state-of-the-art in this domain.

We change the construction of this manuscript and emphasize the innovation of LPPCAP to further demonstrate the insights of this manuscript.

(3) In this paper, the authors proposed a CP-ABE based PIR by utilizing CP-ABE technology. However, the authors do not give the related works about CP-ABE.

We add some CP-ABE and PIR algorithms used in LBS in the related work to demonstrate these schemes can be used in LBS. However the quality of service of these existing schemes is weaker than LPPCAP and we demonstrate this in the experiment result.

(4) The authors proposed a CP-ABE based PIR. However, they do not give the concrete name of the related algorithms.

We change the proposed scheme into LPPCAP and then modify the whole manuscript.

(5) The proposed CP-ABE based PIR is not complete. Especially, key generation algorithm is not given. How can you encrypt and decrypt?

We add the key generation algorithm in 3.2.

Reviewer #2: 

1.It is not clear, what is the role of f(x) in algorithm 1?

We delete the f(x) in algorithm 1.

2.No threat model is presented in the paper. It will be good to include the threat model and present the security analysis accordingly.

We add the threat model in 2.1, and formulate it with probabilityand then present the security analysis in 4.1.

3.It is not clear whether the LBS will know who is sending the query or not? If the LBS knows the individual who sends the query then privacy is lost.

We add the process of how the LBS server know the query and location of the user in the threat model in 2.1, in this part, the way of privacy leakage of user is given.

4. Decryption proof of correctness may be with all steps involved. It is not clear how the second step is derived from the first step.

We add the precise step in algorithm 4, so as to give a clear steps for decryption.

5. Zero Knowledge conceal is used in table 2 however no where it is described.

We add the description of Zero Knowledge leakage about our scheme below table 2. 

6. Figure 8 should be rechecked. X-axis is not clear. Is it time or the number of queries? Similarly figure 6.

The X-axis of figure 8 is queries, and we change all figures in the whole manuscript. 

7. The novelty should be clearly presented.

We change the introduction as well as the abstract to further emphasize the novelty of this manuscript.

Reviewer #3: In this manuscript, in this paper, based on the conception of ciphertext policy attribute-based encryption (CP-ABE), a PIR scheme based on CP-ABE is proposed for preserving the personal privacy in LBS. Personal privacy in LBS is a topic worth studying. However, there are some issues to consider:

1. In the “Introduction”, the authors described that “the superiority of this scheme can be reflected in both privacy preservation and quality of service”. However, the quality of service is not represented in the scheme, if the quality of service is the innovation point, it should be described in detail in the scheme, and it needs to be reflected in the abstract and contribution.

The quality of service of LPPCAP is a part of innovation, so we add the reflection of this point in both the abstract and contribution.

2. The “3.1 The conception of CP-ABE based PIR” should describ in the “Preliminaries”, and the system architecture and privacy threats are more appropriate in the “CP-ABE based PIR”, the front of the scheme description.

We change the conception of LPPCAP in the “Preliminaries”, and then change the system architecture and the threat model to further appropriate to the conception of LPPCAP in 2.3.

3. What is the relationship between this scheme and literature 8, and what are the advantages of this scheme compared with literature 8?

The basic idea and conception of our proposed LPPCAP is derived from literature 8, however, in literature 8, the proposed scheme is mainly in theoretical and without any practice. So in this manuscript,we attempt to utilize the theoretical result in LBS to further the utilization of protocol proposed in literature. In addition, LPPCAP can be seen as a practical of the theory produced in literature [8]. We add the description of above in 4.1.

---

## [Decision Letter · Decision Letter 1]

14 Aug 2024

PONE-D-24-26572R1A location-based service scheme with attribute information privacyPLOS ONE

Dear Dr. Dai,

Thank you for submitting your manuscript to PLOS ONE. After careful consideration, we feel that it has merit but does not fully meet PLOS ONE’s publication criteria as it currently stands. Therefore, we invite you to submit a revised version of the manuscript that addresses the points raised during the review process.

We look forward to receiving your revised manuscript.

Kind regards,

Zhiquan Liu, Ph.D.

Academic Editor

PLOS ONE

Journal Requirements:

Additional Editor Comments:Almost all reviewers are very positive about the contribution of this paper, and also point out some constructive comments. Please revise as much as possible according to the opinions as soon as possible, and submit the revised version to ensure that this paper can be published quickly.

Reviewers' comments:

Reviewer's Responses to Questions

**Comments to the Author**

1. If the authors have adequately addressed your comments raised in a previous round of review and you feel that this manuscript is now acceptable for publication, you may indicate that here to bypass the “Comments to the Author” section, enter your conflict of interest statement in the “Confidential to Editor” section, and submit your "Accept" recommendation.

Reviewer #2: All comments have been addressed

Reviewer #4: (No Response)

2. Is the manuscript technically sound, and do the data support the conclusions?

Reviewer #2: Partly

Reviewer #4: Yes

3. Has the statistical analysis been performed appropriately and rigorously? 

Reviewer #2: I Don't Know

Reviewer #4: Yes

4. Have the authors made all data underlying the findings in their manuscript fully available?

Reviewer #2: Yes

Reviewer #4: Yes

5. Is the manuscript presented in an intelligible fashion and written in standard English?

Reviewer #2: Yes

Reviewer #4: Yes

6. Review Comments to the Author

Reviewer #2: The grammatical error in the revision part (in yellow color) has to be rechecked. The sentences are not complete or not conveying the meaning.

Reviewer #4: In this scheme, query and feedback are encrypted with security two-parties calculation by the user and the LBS server, so as not to violate any personal privacy and decrease the processing time in encrypting the retrieved information. In addition, this scheme can also preserve the attribute privacy of users such as the query frequency as well as the moving manner. This article is well written, I think it can be accepted, while the following comments should be revised.

(1) The title of the picture is recommended to be placed below the picture, not above it. (2) In the figure of experimental results, it is suggested to add references for comparison schemes. (3) A related work, fct: a fully-distributed context-aware trust model for location based service recommendation, is suggested to be discussed or compared in related work. (4) It is recommended that the serial number before Introduction be changed to 1.

7. PLOS authors have the option to publish the peer review history of their article (what does this mean?). If published, this will include your full peer review and any attached files.

Reviewer #2: No

Reviewer #4: No

---

## [Author Response · Author response to Decision Letter 1]

19 Aug 2024

Response to reviewer's comments

Reviewer #2: The grammatical error in the revision part (in yellow color) has to be rechecked. The sentences are not complete or not conveying the meaning.

We rechecked the revision part (in yellow color) and changed these sentences.

Reviewer #4: In this scheme, query and feedback are encrypted with security two-parties calculation by the user and the LBS server, so as not to violate any personal privacy and decrease the processing time in encrypting the retrieved information. In addition, this scheme can also preserve the attribute privacy of users such as the query frequency as well as the moving manner. This article is well written, I think it can be accepted, while the following comments should be revised.

(1) The title of the picture is recommended to be placed below the picture, not above it. (2) In the figure of experimental results, it is suggested to add references for comparison schemes. (3) A related work, fct: a fully-distributed context-aware trust model for location based service recommendation, is suggested to be discussed or compared in related work. (4) It is recommended that the serial number before Introduction be changed to 1.

(1)We change the title of the picture to the place below the picture. 

(2)The references are added in figure of experimental results.

(3)A related work is discussed in related work, and we added it as “Liu et al. [30] also proposed a fully-distributed context-aware trust model for location based service. ” in section 2.

(4)The serial number before Introduction is changed to 1.

---

## [Editor Report · Decision Letter 2]

21 Aug 2024

A location-based service scheme with attribute information privacy

PONE-D-24-26572R2

Dear Dr. Dai,

We’re pleased to inform you that your manuscript has been judged scientifically suitable for publication and will be formally accepted for publication once it meets all outstanding technical requirements.

Kind regards,

Prof. Zhiquan Liu

Academic Editor

Jinan University

zqliu@vip.qq.com

https://www.zqliu.com

Additional Editor Comments (optional):

Accept
---

## [Editor Report · Acceptance letter]

28 Aug 2024

PONE-D-24-26572R2 

PLOS ONE

Dear Dr. Dai, 

I'm pleased to inform you that your manuscript has been deemed suitable for publication in PLOS ONE. Congratulations! Your manuscript is now being handed over to our production team.

Kind regards, 

on behalf of

Professor Zhiquan Liu 

Academic Editor

PLOS ONE